# Exogenous Application of Glycine Betaine on Sweet Cherry Tree (*Prunus avium* L.): Effects on Tree Physiology and Leaf Properties

**DOI:** 10.3390/plants11243470

**Published:** 2022-12-11

**Authors:** Marta Serapicos, Sílvia Afonso, Berta Gonçalves, Ana Paula Silva

**Affiliations:** 1Department of Agronomy, University of Trás-os-Montes e Alto Douro (UTAD), 5000-801 Vila Real, Portugal; 2Centre for the Research and Technology of Agro-Environmental and Biological Sciences (CITAB), Institute for Innovation, Capacity Building and Sustainability of Agri-Food Production (Inov4Agro), University of Trás-os-Montes e Alto Douro (UTAD), 5000-801 Vila Real, Portugal; 3Department of Biology and Environment, University of Trás-os-Montes e Alto Douro (UTAD), 5000-801 Vila Real, Portugal

**Keywords:** sweet cherry, biostimulants, glycine betaine, plant physiology, leaf compounds

## Abstract

Biostimulants, such as glycine betaine (GB), are a sustainable way to boost productivity and quality in fruit crops, even in adverse environment conditions. Sweet cherry (*Prunus avium* L.) is an important crop, which is very sensitive to abiotic stress. Known primarily for its fruits, its leaves are also rich in bioactive substances, which, however, still have no commercial value. There are no studies about the effects of GB exogenous application on biochemical parameters of sweet cherry leaves and few studies about effects in sweet cherry tree physiology. This study was conducted in a Portuguese sweet cherry commercial orchard. *Lapins* sweet cherry trees were treated with a commercial product based on GB, at two different concentrations (0.25% and 0.40%). The applications were performed at three different phenological stages, according to the BBCH (*Biologische Bundesanstalt, Bundessortenamt und Chemische Industrie*) scale: 77, 81 and 86 BBCH. Both GB treatments improved leaf water status, photosynthetic pigments, soluble solids content, total phenolic contents, and antioxidant activity, resulting in better leaf water status regulation, greater photosynthetic capacity, and higher antioxidant activity. These results shows that GB can benefit sweet cherry tree physiology and provide new opportunities for sweet cherry leaves valorisation.

## 1. Introduction

Plant biostimulants have grown in popularity because of their ability to combine the concepts of productivity, quality, and sustainability in agricultural production [1]. Glycine betaine (GB) is recognized as a biostimulant capable of combating abiotic stress [2]. As a secondary metabolite, it is synthesized and accumulated primarily in higher plants for osmotic regulation. Because of its osmoprotective and osmoregulatory properties it is an intriguing compound to apply exogenously to crops to increase yield and quality, sustainably [3,4,5]. The sweet cherry tree (*Prunus avium* L.) is one of the few cultivated Rosaceae species of the subgenus *Cerasus* [6]. This tree is best known for its sweet cherries, also known as cherries [7]. It is grown in temperate climates and has a significant social and economic impact in high-production areas [8]. Because cherry tree yield is easily affected by adverse weather conditions, the use of biostimulants has been a very appealing option for improving tree resistance and increasing productivity even in adverse environments [9].

Sweet cherries are rich in health-promoting compounds [7,10]. Aside from its fruits, this tree provides other parts such as sweet cherry peduncles (stems), seeds, and leaves. These parts have been used for medicinal and well-being purposes since ancient times, but they have no commercial application or significance today. However, recent studies have shown that seeds and peduncles have the potential to become high-value bioproducts [11,12]. Peduncles even contain more phenolic compounds and have a higher antioxidant potential than fruits [12,13]. The few studies on sweet cherry leaves found in the literature revealed that they are also full of bioactive compounds and, as peduncles, have higher concentrations of phenolic compounds and superior antioxidant activity than the fruits [14,15].

Even though GB is a well-studied compound, there have been few studies on the effects of its exogenous application in sweet cherry tree physiology. Furthermore, no studies on the effects of this biostimulant in leaf compounds were found. In this context, the purpose of this study is to assess the effects of exogenous GB application on the physiology of the sweet cherry tree as well as the biochemistry of leaves.

## 2. Results and Discussion

### 2.1. Leaf Water Status

Sweet cherry leaf water status results are presented in Figure 1.

Regarding relative water content (RWC) results (Figure 1A), unlike the first sampling of leaves, the second sampling revealed significant differences among treatments (*p <* 0.05). In the first sampling, RWC ranged between 88.73% and 89.97%. There was an increase in RWC in the second sampling for all treatments. GB 0.40%, GB 0.25%, and control treatment led to RWC values of 93.50, 92.97, and 91.99%, respectively. Results show higher RWC in leaves of trees treated with GB; however, it was the GB 0.40% treatment that induced better water status. Higher RWC values are related to proper water uptake, greater capacity to hold water inside cells, and may even reveal low water stress levels [16]. Furthermore, RWC levels are related, although indirectly, to improved metabolic processes, namely photosynthetic efficiency, once RWC also influences cell turgor, stomata movement, carbon dioxide intake, and consequently photosynthetic rate [17]. Similar results were obtained in experiments with other species, namely *Carica papaya* L. cv. *BH-65* [18]; *Lycopersicon esculentum* cv. *L00587*, *L0060* [19], *Marwa hibrid* [20], and *PS* [21]; *Malus hupehensis* Rehd. cv. *Pinyiensis Jiang* [22]; *Olea europaea* L. cv. *Koroneiki* [23], *Chondrolia,* and *Chalkidikis* [24]. Correia et al., (2020) [25] also reported a RWC increase in sweet cherry cv. *Skeena* after GB spraying at preharvest.

Leaf mass per unit area (LMA) (Figure 1B) and cuticular transpiration rate (CTR) (Figure 1C) values also increased from first to second leaf sampling because of increasing leaf age as well as due to relatively higher mean air temperatures during June (Appendix A). Although LMA and CTR results do not present significant differences among treatments, the values corroborate RWC findings. Leaf mass per unit area values are higher in GB-treated leaves, which means that GB can lead to denser or/and thicker leaves [26]. Higher LMA values are correlated with improvements in leaf gas exchange, leaf toughness, and relative growth rate [27,28]. Cuticular transpiration rate values are lower in GB-treated leaves, which means that GB can limit CTR values and help maintain water inside leaf tissues.

### 2.2. Electrolyte leakage and thiobarbituric acid reactive substances

Figure 2 shows the values of electrolyte leakage (EL), and thiobarbituric acid reactive substances (TBARS) of the two samples of cherry leaves of the cv. *Lapins* according to the different treatments.

Electrolyte leakage (Figure 2A) and TBARS (Figure 2B) analysis results of the first leaf sampling showed no significant differences between treatments (*p* > 0.05). The second leaf sampling, EL and TBARS values presented significant differences between treatments (*p <* 0.001). Electrolyte leakage values range from 17.5 to 25.3%, and TBARS ranges from 538 to 699 nmol g^−1^. In both methods, GB increased values when compared to control trees, with the highest ones coming from leaves treated with GB 0.40%.

Plant cell membranes have selective permeability. Excessive reactive oxygen species (ROS) accumulation, due to biotic or abiotic stresses, can damage cell membranes through lipid peroxidation and change their permeability [29]. This will cause a loss of electrolytes to the extracellular medium, an increase of lipid peroxidation by-products, and a general decrease in cell tolerance to adverse conditions. Thus, an increase in the concentration of electrolytes and TBARS translates into cell membrane deterioration [30]. In this way, higher EL and TBARS concentrations in leaves treated with GB suggest that this osmoprotectant did not prevent membrane lipid peroxidation, allowing the release of electrolytes into the extracellular medium. Denaxa et al., (2020) [31], reported that GB application also did not lead to improvements in terms of membrane lipid peroxidation nor EL in *Olea europaea* L. cv. *Koroneiki*. However, in similar studies with *Lycopersicon esculentum* and *Zea mays* L., GB exogenous application reduced oxidative damage and EL [19,32].

### 2.3. Total Phenolics and Antioxidant Activity

#### Total Phenolics and Antioxidant Activity

Results of the total phenolics and antioxidant activity, by the methods 2,2-azinobis(3-ethylbenzothiazoline-acidosulfonic) acid (ABTS), 2,2-diphenyl-1-picrylhydrazil (DPPH), and *β*-Carotene of the two sampling dates of sweet cherry leaves of the cultivar *Lapins* subjected to the different treatments can be observed in Figure 3.

For the total phenolic content (Figure 3A), there were significant differences (*p <* 0.05) for the first sampling of leaves and for the second sampling (*p <* 0.001), between the treatments applied. In the first sampling of leaves, the total phenolic content varies between 29.07 and 37.23 mg gallic acid equivalents (GAE) g^−1^ DW. The concentration of these compounds increased with leaf age [33]; therefore, analysis of the second sampling revealed higher total phenolic contents, with a range between 53.91 and 72.07 mg GAE g^−1^ DW. On both sampling dates, the highest value of total phenolics corresponds to the tree’s leaves treated with GB 0.40%. Regarding the antioxidant activity, in the same way as with the total phenolics, there was also a general increase in the antioxidant activity from the first sampling to the second sampling. In both samplings, the ABTS method (Figure 3B) did not reveal significant differences between treatments (*p* > 0.05). Leaf antioxidant capacity by the DPPH (Figure 3C) and *β*-Carotene (Figure 3D) methods showed significant differences (*p <* 0.001) between treatments in both samplings. In the first sampling, the DPPH method results ranged between 77.54 and 95.01 µmol trolox equivalents (TE) g^−1^ DW and in the second one varied between 84.43 and 108.98 µmol TE g^−1^ DW. In this same method, the exogenous application of GB led to an increase in the antioxidant activity in the leaves, which is related to the greater accumulation of total phenolics. *β*-Carotene inhibition values in the first sampling ranged from 81.45 to 88.37% and in the second sampling from 86.77 and 96.25%. In this case, the second sampling followed the tendency of the DPPH method; however, in the first sampling the percentage of inhibition was lower in the leaves of the trees treated with GB 0.40% and higher in the leaves treated with GB 0.25%. In general, foliar exogenous application of GB increased the concentration of the total phenolic, and this event was further evidenced by treatment with GB 0.40%, leading to higher antioxidant capacity. In other studies, namely with Chinese crab apple [22], mango [34], and olive tree [31] the application of GB also led to the accumulation of phenolic compounds in the leaf and consequently to greater antioxidant activity. Phenolics biosynthesis may be related to the presence of ROS that can cause EL and trigger antioxidant mechanisms [35]. In this way, higher total phenolics in leaves of plants treated with GB may also be linked to higher EL and TBARS concentrations.

### 2.4. Photosynthetic Pigments

Figure 4 shows the leaf photosynthetic pigments, chlorophyll a (Chl_a_), chlorophyll b (Chl_b_), total chlorophyll (Chl_total_), and total carotenoids (Crtn_total_) concentrations and the Chl_a_/Chl_b_ and Chl_total_/Crtn_total_ ratios for the two sampling dates according to the different treatments.

Regarding the first sampling of leaves, GB treatments affected the Chl_b_ (Figure 4B) concentration (*p <* 0.05), without affecting Chl_a_ (Figure 4A), Chl_total_ (Figure 4C), Crtn_total_, (Figure 4E) Chl_a_/Chl_b_ (Figure 4D), and Chl_total_/Crtn_total_ (Figure 4F). Leaf pigments analysis from the second sampling showed significant differences for Chl_a_, Chl_b_, Chl_total_, and Chl_a_/Chl_b_ (*p* < 0.05); for Crtn_total_ (*p <* 0.01); and no differences (*p* > 0.05) for Chl_total_/Crtn_total_ ratio between treatments. The photosynthetic pigment concentration increases with leaf age until the maximum value is reached with maturation [36]. Likewise, there was an increase in photosynthetic pigment concentration from the first to the second leaf sampling. Chlorophyl a is in the reaction centres and in the light-harvesting complexes, being the most abundant pigment in this study and in the majority of terrestrial plants [37]. In the second sampling, the values of Chl_a_, Chl_b_, Chl_total_, and Crtn_total_ varied between 2.35–2.90 mg g^−1^ dry weight (DW), 1.36–1.66 mg g^−1^ DW, 3.71–4.56, and 0.49–0.59 mg g^−1^ DW, respectively. Both GB exogenous foliar treatments led to increases in chlorophyll concentration and Crtn_total,_ yet the higher concentration of GB led to superior increases. Respiration and photosynthesis rates are directly and positively correlated with pigment concentration, whether of chlorophylls or carotenoids [38,39]. In addition, they are endowed with high antioxidant capacity, which agrees with the antioxidant activity results of sweet cherry leaves, through the DPPH and *β*-Carotene methods, obtained in this study [40,41]. In this way, GB accumulation may protect photosynthetic pigment degradation or activate photosynthetic pigment biosynthesis, contributing to better photosynthetic rates and higher antioxidant activity [25]. In other studies with sweet cherry [25], as well as other fruit plants, such as hibiscus [42], mango [34], olive tree [23,24], strawberry [43,44], and tomato [19,20,45], GB applications also led to improvements in photosynthetic pigment content.

The ratio Chl_a_/Chl_b_ increased with the GB 0.25% treatment in the second sampling of leaves. This ratio depends essentially on genetic and environmental factors [46]. As higher Chl_a_/Chl_b_ values are related to a greater capacity to absorb light and consequently higher photosynthetic rates, GB treatments may have improved plant photosynthetic capabilities [47]. Though there are no significant differences regarding Chl_total_/Crtn_total_ ratio, different tendencies were revealed in the two samplings. In the first sampling, Chl_total_/Crtn_total_ analysis revealed lower values for GB-treated leaves while in the second sampling the opposite occurred, as Chl_b_ carotenoids are also considered accessory pigments but can harvest wide ranges of sunlight wavelengths. Additionally, when the incident radiation exceeds the amount needed for photosynthesis, carotenoids accept energy from chlorophyll and dissipate that energy as heat, preventing chloroplast and tissue damage [37].

### 2.5. Soluble Sugars, Starch, Soluble Proteins, and Proline Content

Soluble sugars, starch, soluble proteins, and proline concentrations of cv. *Lapins* leaves subjected to the different treatments are presented in Figure 5.

For soluble sugars and starch contents, in both analyses of leaves from the first and second samplings there are significant differences between treatments (*p <* 0.001). Regarding soluble proteins, there are no significant differences (*p* > 0.05). Proline content results show significant differences for the first sampling (*p* < 0.05) between treatments and no differences in the second sampling (*p* > 0.05).

Soluble sugars concentration (Figure 5A) ranged from 133.61 to 248.02 mg glucose (Glc) g^−1^ DW, and 96.58 to 122.62 mg Glc g^−1^ DW, in the first and second sampling, respectively. In turn, starch content (Figure 5B) ranged from 104.20 to 143.91 mg Glc g^−1^ DW and 120.80 to 171.25 mg Glc g^−1^ DW. From the first to the second sampling soluble sugar contents decreased because of mobilization of soluble sugars to other developing organs, namely fruits. For starch, the inverse followed, as periods of active photosynthesis produce excess of carbohydrates, which are stored in leaves as starch for future remobilizations [16,48]. In both samplings there occurred higher values of soluble sugars and starch content in leaves treated with GB. These results are supported by Correia et al., (2020) [25], who carried out an experiment with sweet cherry cv. *Skeena*. Proline content decreased from the first to second sampling. Glycine betaine treatments led to significantly higher proline contents (Figure 5D) in the first sampling and tendentially higher contents in the second sampling. As mentioned above, GB may have led to improvements in leaf water status, photosynthetic pigments, and photosynthetic rates, which could explain increased biosynthesis and accumulation of carbohydrates and proline in GB-treated leaves [48]. Soluble proteins content (Figure 5C) was reduced between the first and second sampling by GB exogenous application, although there are no significant differences between treatments. As in this study, EL and TBARS increased in leaves treated with GB, which is linked to cell membrane damage, and the occurrence of a slight decrease in the content of soluble proteins is coherent [49].

## 3. Materials and Methods

### 3.1. Experimental Design and Plant Material

The experiment was carried out in 2021 in a 7-year-old sweet cherry (*Prunus avium* L.) orchard in the north of Portugal, at Quinta da Alufinha, in the municipality of Resende (latitude 41°06′ N and longitude 7°54′ W), planted at an elevation of 140 m above sea level. The orchard has a drip irrigation system which ensures a uniform supply of water. According to the Köppen climate classification, Resende has a Mediterranean warm summer climate, with an average annual temperature of 12.4 °C. The average annual rainfall is 1299 mm, with the driest month being July, with an average of 27 mm. Although this orchard has several cultivars, for reasons of consistency of results this study focused only on the cultivar *Lapins*, also called *Cherokee*, grafted rootstock Santa Lucia 64, characterized for being late and very productive [6,50].

Three treatments were used in this study, all of which were applied via foliar application. In two of the treatments, a commercial product based on GB (97%) was used in two concentrations: 0.25% (GB 0.25%) and 0.40% (GB 0.40%). Water was used as the control treatment for the third treatment. Each treatment had eight trees that were homogeneous and in good phytosanitary conditions, for a total of 24 trees. Figure 6 depicts the location of the trees (represented by the lines) and the treatments (represented by the rectangles). Since the commercial orchard was very homogeneous in terms of soil type, slope and selected trees, each tree was assumed to be a repetition.

All treatments were repeated on three dates at different phenological stages according to the BBCH (*Biologische Bundesanstalt, Bundessortenamt und Chemische Industrie*) scale: 77 BBCH (fruit about 70% of the final size) (Figure 7A), 81 BBCH (beginning of fruit colouring) (Figure 7B) and 86 BBCH (colouring advanced) (Figure 7C) (three days before harvest). Treatments were foliar sprays with a mechanical sprayer when there was no rain forecast within the next 24 h.

At least 64 adult healthy leaves per treatment (eight per tree) were collected after the first (first sampling, 7 May 2021) and final (second sampling, 7 June 2021) GB treatments. The leaves were divided into three groups in the field, packaged in the cold, and immediately transported to the laboratory. The first group was used for measurements relating to the leaf water status. The second group was used for EL analysis. The third group of leaves was used for biochemical analyses. This last group was lyophilized for one week at −50 °C in a SCANVAC CoolSafe lyophilizer (LABOGENE, Lillerød, Denmark). The lyophilized leaves were then ground and sealed in bags

### 3.2. Physiological Parameters Determination

#### 3.2.1. Leaf Water Status

Relative water content is a good indicator of plant water status because it translates the amount of absolute water that plant tissues can absorb until reaching the state of turgor [51]. In this way, first, a representative leaf of each tree was precisely detached and placed inside a falcon tube with the petiole facing downwards. Leaf fresh weight (*FW*) in g was measured. Then the leaf petiole was submerged in deionized water, and the falcon tubes were stored in a dark place at 4 °C. After 24 h, turgid weight (*TW* in g) was measured. Then, the leaf was dried at 70 °C until a constant weight was reached. Leaf dry weight was weighed in g. Relative water content in % was obtained according to the formula: $RWC\;\left(\%\right)=\frac{FW-DW}{TW-DW}\times 100$. Leaf mass per unit area (in g m^−2^) was calculated by the ratio between leaf area (*Area_leaf_* in m^2^) and *DW*. Cuticular transpiration rate was also calculated, using the formula $CTR=\frac{FW-DW}{Area_{leaf}}$. All the values are presented as the mean ± standard deviation (SD) of eight replicates.

#### 3.2.2. Electrolyte Leakage

Electrolyte leakage from the leaf tissues was determined according to the method of [52]. One leaf was collected, washed with ultrapure water to remove electrolytes from the surface, and a disk-shaped segment was cut and placed in a falcon tube with 10 mL of ultrapure water. Incubation at 25 °C was carried out for 24 h, and the electrical conductivity of the solution was measured ($C_{E1}$). Then, the samples were placed in an autoclave at 120 °C for 20 min, cooled to 25 °C, and the electrical conductivity of the solution was measured again ($C_{E2}$). The following formula was used to determine *EL* from plant tissues: $EL\;\left(\%\right)=\frac{C_{E1}}{C_{E2}}\times 100$. The values are presented as the mean ± SD of eight replicates.

#### 3.2.3. Thiobarbituric Acid Reactive Substances

Cell membrane lipid peroxidation was determined according to [53,54]. First, 4 g of lyophilized sample leaves were placed in plastic centrifuge tubes, followed by 6 mL of 20% trichloroacetic acid, vortexed, and centrifuged at 4000 rpm for 20 min at 4 °C. In a glass tube, 1 mL of extract, 1 mL of 0.5% thiobarbituric acid (TBAR), and 100 mL of butyl hydroxytoluene were placed. These tubes were vortexed and placed in a 95 °C open water bath for 30 min. The tubes were then centrifuged for 20 min at 3000 rpm and 4 °C. Supernatant absorbance (Abs) was measured at 532 nm and 600 nm. Total thiobarbituric acid reactive substances concentration was expressed as nmol g^−1^ DW, using an extinction coefficient of 155 M cm^−1^, as the mean ± SD of eight replicates.

### 3.3. Biochemical Compounds Analysis

#### 3.3.1. Leaf Sample Extraction Procedure

To analyse leaves’ antioxidant activity and phenolic compounds, their extracts were prepared in advance. Three leaves per tree were collected, lyophilized, and ground. Forty mg DW of each sample were weighed into an Eppendorf tube with 1.5 mL of methanol 70%. Tubes were mixed thoroughly in a vortex for 30 min and centrifuged at 5000 rpm at 4 °C for 15 min. The supernatant was collected in a 10 mL volumetric flask. The process was repeated three times. After the supernatants from the three extractions were put into volumetric flasks, the volume was made up with methanol. Extracts were stored in Falcon tubes at −20 °C until further analysis.

#### 3.3.2. Leaf Total Phenolics

Total phenolic content was determined using the spectrophotometry method developed by Singleton and Rossi (1965) [55] and Dewanto et al., (2002) [56]. In each well of a 96-well microplate, 20 µL of each leaf extract sample, 100 µL of Folin-Ciocalteau reagent (1:10 in bidistilled H_2_O), and 80 µL of 7.5% Na_2_CO_3_ were mixed. The microplate was then incubated in the dark for 15 min at 45 °C. After incubation, absorbance (Abs) values were measured against a blank in a SPECTROstarNano microplate reader at 765 nm (BMG LABTECH, Ortenberg, Germany). The values were calculated as the mean ± SD of eight replicates using a calibration curve of gallic acid standard solution and expressed as GAE g^−1^ DW.

#### 3.3.3. Leaf Antioxidant Activity

To assess antioxidant activity, ABTS, DPPH, and β-Carotene linoleic acid bleaching assay were used.

##### ABTS Method

Antioxidant capacity of sweet cherry extract was determined using an ABTS assay, described by Re et al., (1999) [57]. A previously prepared ABTS radical solution was diluted with ethanol to lead to an Abs of 0.70 ± 0.02 units at 734 nm. In each well of a 96-well microplate, 15 mL of extract was mixed with 285 mL of ABTS solution. For 10 min, the microplates were placed in a dark place. Absorbance was then measured in a microplate reader at 734 nm. Results were expressed as g of Trolox equivalents (TE) g^−1^ DW as the mean ± SD of eight replicates using a calibration curve of Trolox.

##### DPPH Method

A spectrophotometric method adapted from Siddhuraju and Becker (2003) [58] was used to assess the antioxidant capacity by DPPH radical scavenging. First, 15 µL of extract and 285 µL of freshly prepared DPPH solution were added to each 96-well microplate well. The microplates were left at room temperature in a dark place for 30 min. The Abs values were measured at 517 nm, and results were expressed as mol TE g^−1^ DW as the mean ± SD of eight replicates from a Trolox calibration curve.

##### β-Carotene-Linoleic Acid Bleaching Assay

A β-carotene-linoleic acid bleaching assay was carried out using a method developed by Salleh et al., (2012) [59]. In a round-bottomed flask, 0.5 mg β-carotene, 1 mL chloroform high-performance liquid chromatography grade, 25 µL linoleic acid, and 200 mg Tween 40 were mixed. Following that, chloroform was completely evaporated under vacuum at 40 °C, and 100 mL of distilled water was added to the residue. The mixture was gently whisked together until it formed a clear yellowish emulsion. Then, in a 96-well microplate, 50 µL of the extract was added to each well, along with 0.25 mL of the yellowish emulsion. Absorbance against the blank was then measured at 470 nm, and the microplate was incubated in a hot plate, light-protected, at 50 °C. After 2 h, Abs was measured again at 470 nm. The results were expressed as the extract’s β-Carotene inhibition percentage, which was calculated as follows: I% = (Abs_β-carotene after 2 h/_Abs_initial β-carotene_) × 100, where Abs_β-carotene after 2 h_ is the Abs value after 2 h of incubation and Abs_initial β-carotene_ the Abs value before incubation. All results were presented as the mean ± SD of eight replicates.

#### 3.3.4. Photosynthetic Pigments

For the determination of photosynthetic pigments, a spectrophotometric method based on Lichtenthaler (1987) was used [60]. First, 25 mg of DW was weighed into a 10 mL screw tube, and 4 mL of 80% acetone was added. The mix was then homogenized in a vortex, sonicated for 5 min at 30 Hz, and centrifugated at 4000 rpm at 4 °C for 10 min. Then, 200 µL of each sample was transferred into a 96-well microplate, and Abs was measured in a microplate reader at 470, 645, and 663 nm against a blank. Absorbance values allowed the determination of different photosynthetic pigments: Chl_a_, Chl_b_, and Crtn_total_. Additionally, Chl_total_, ratio Chl_a_/Chl_b_ as well as Chl_total_/Crtn_total_ were also determined. All the results were expressed as mg g^−1^ DW, as the mean ± SD of eight replicates.

#### 3.3.5. Soluble Sugars and Starch Content

Soluble sugars and starch contents were measured by the spectrophotometry method according to Ni et al., (2009) [61]. First, leaf extracts were prepared by adding 10 to 40 g of lyophilized ground samples into Eppendorf centrifuge tubes. Two mL of 100% acetone was added, and the tubes were placed in the sonicator for 5 min. Extracts were then centrifuged at 11,000 rpm at 4 °C for 10 min, and the supernatant was discarded (residue was preserved). To determine soluble sugars, 2.5 mL of 80% ethanol was added to the residue and the tubes were placed at 80 °C for 30 min. Tubes were then centrifuged at 11,000 rpm at 4 °C for 10 min, and the supernatant was collected to determine starch content using spectrophotometry. To determine starch content, the supernatant of soluble sugars’ extraction was discarded, 2 mL of 1.1% HCl was added to the respective residue, and the tubes were placed at 100 °C. After 30 min, the mixture was cooled in ice, and the supernatant was collected to determine starch content using spectrophotometry. In both procedures, 100 µL of extract and 500 µL of anthrone 0.2% were placed in Eppendorf tubes for spectrophotometric analysis. After that, the tubes were heated at 100 °C for 10 min before being quickly cooled to 0 °C. Finally, 200 µL of each supernatant sample was pipetted to a 96-well microplate, and Abs values were measured against a blank at 625 nm. The values were quantified using a calibration curve of Glc as mg Glc g^−1^ DW, as the mean ± SD of eight replicates.

#### 3.3.6. Total Soluble Proteins

Total soluble proteins were extracted using 1400 µL of an extraction buffer containing phosphate buffer (pH 7.5), 0.1 mM ethylenediaminetetraacetic acid (EDTA), 100 mM phenyl-methylsulfonyl fluoride (PMSF), and 2% (*w*/*v*) polyvinylpyrollidone (PVP), followed by centrifugation at 12,000× *g* at 4 °C for 30 min. Absorbance was read at 595 nm, and bovine serum albumin (BSA) was used as a standard [62]. The results were expressed as mg g^−1^ DW, as the mean ± SD of eight replicates.

#### 3.3.7. Free Proline Content

Proline content of cherry leaves was determined according to the method of Bates et al., 1973, Bernardo et al., 2017, and Carvalho et al., (2019) [63,64,65]. First, 20 mg of lyophilized sheet sample was weighed into an Eppendorf tube, 1.5 mL of 3% (*w*/*v*) sulphosalicylic acid was added, and the tubes were centrifuged for 15 min at 12,000 rpm at 4 °C. The mixture was then heated on a water bath at 100 °C with 0.2 mL of supernatant, 0.2 mL of acid-ninhydrin, and 0.2 mL of glacial acetic acid in another Eppendorf tube. After 1 h, the tubes were rapidly cooled on ice, 0.8 mL of toluene was added, and the Eppendorf tubes were vortexed. After the tubes had reached room temperature, 200 µL of each sample was transferred into a 96-well microplate and the Abs at 520 nm was measured in a microplate reader against a blank. Free proline content was estimated by interpolation with a standard curve of L-proline: $\left[proline\;\frac{µmol}{g\;sample}\right]=\frac{\left[\left(\frac{µg\;proline}{ml}\right)\times 0.8\;\left(ml\;toluene\right)\times 1.5\;\left(\;ml\;sulphosalicylic\;acid\right)\right]}{\left[115.13\;\frac{µg}{µmole}\times sample\;\left(g\right)\right]}$. The results were expressed as µmol proline g^−1^ DW, as the mean ± SD of eight replicates.

### 3.4. Statistical Analysis

Statistical Package for Social Science (SPSS) software, version 27.0, was used for the statistical analyses (IBM Corporation, New York, NY, USA). One-way ANOVA was used to analyse the dependent variables. When the assumptions of normality and variance homogeneity were not met, the Welch correction was used, and when these were verified, Tukey’s post-hoc (HSD) test was used. When a significant effect (*p* < 0.05) was observed in the test with Welch correction, the Dunnett T3’s test was used. Because the number of samples was less than 50, the Shapiro-Wilk tests were used to assess the assumption of normal distribution, and the Levene test was used to assess the assumption of variance homogeneity.

## 4. Conclusions

Glycine betaine application in sweet cherry cv. *Lapins* improved the tree’s physiological and biochemical performance. Although GB 0.40% produced the best results, both GB treatments increased leaf water status, photosynthetic pigment concentration, soluble sugar and starch content, total phenolic content, and leaf antioxidant activity. On the other hand, GB did not prevent membrane degradation since leaves treated with this osmolyte had higher electrolyte and TBARS concentrations. However, the plants had improved water regulation, photosynthesis, biosynthesis of important metabolites, and antioxidant capacity. In addition to beneficial GB effects on sweet cherry tree performance, these results provide new opportunities for the valorisation of sweet cherry leaves as a by-product. Furthermore, GB promises an adaptation of sweet cherry tree to stressful abiotic conditions, such as drought.

## Figures and Tables

**Figure 1 plants-11-03470-f001:**
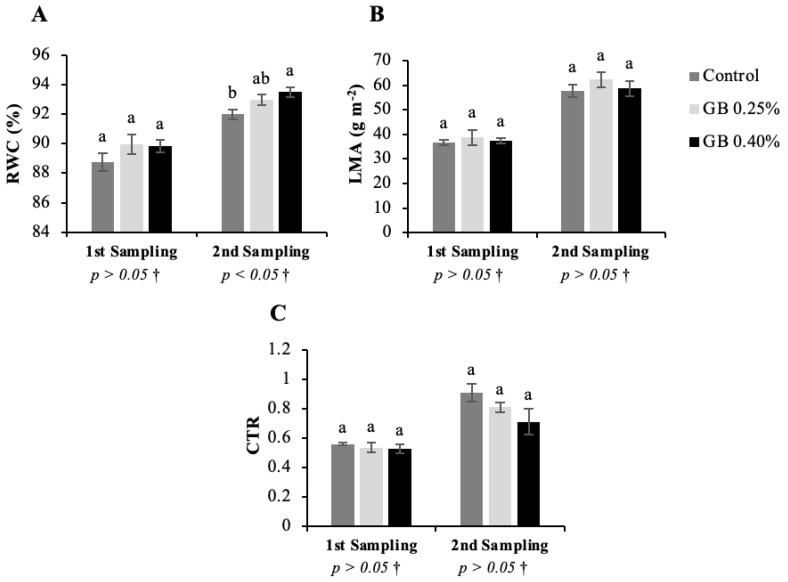
Represents leaf water status in sweet cherry tree leaves. Relative water content (RWC (**A**), leaf mass per ares (LMA) (**B**), and cuticular transpiration rate (CTR) (**C**), for the cv. *Lapins* leaves subjected to the different treatments. † Indicates that the *p* values were calculated from the one-way ANOVA analysis after verifying the assumptions of normality and homogeneity of variances. When a significant effect was observed (*p* < 0.05), Tukey’s post-hoc test was applied.

**Figure 2 plants-11-03470-f002:**
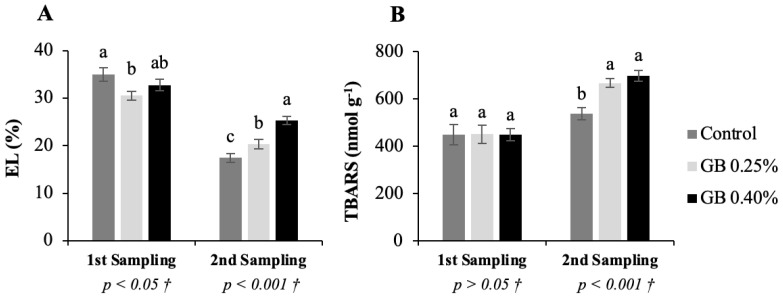
Represents electrolyte leakage (EL) and thiobarbituric acid reactive substances (TBARS) in sweet cherry tree leaves. EL (**A**), and TBARS (**B**), for the cv. *Lapins* leaves subjected to the different treatments. † Indicates that the *p* values were calculated from the one-way ANOVA analysis after verifying the assumptions of normality and homogeneity of variances. When a significant effect was observed (*p* < 0.05), Tukey’s post-hoc test was applied.

**Figure 3 plants-11-03470-f003:**
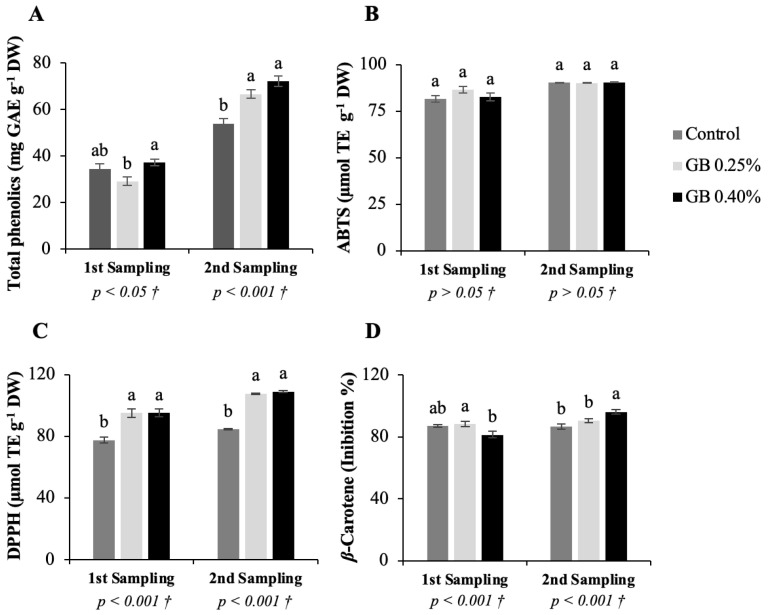
Represents total phenolics and antioxidant activity results in sweet cherry tree leaves. Total phenolic content (**A**) and antioxidant activity by 2,2-azinobis(3-ethylbenzothiazoline-acidosulfonic) acid (ABTS) (**B**), 2,2-diphenyl-1-picrylhydrazil (DPPH) (**C**), and *β*-Carotene (**D**) methods, for the cv. *Lapins* leaves subjected to the different treatments. † Indicates the *p* values were calculated from the one-way ANOVA analysis after verifying the assumptions of normality and homogeneity of variances. When a significant effect was observed (*p* < 0.05), Tukey’s post-hoc test was applied.

**Figure 4 plants-11-03470-f004:**
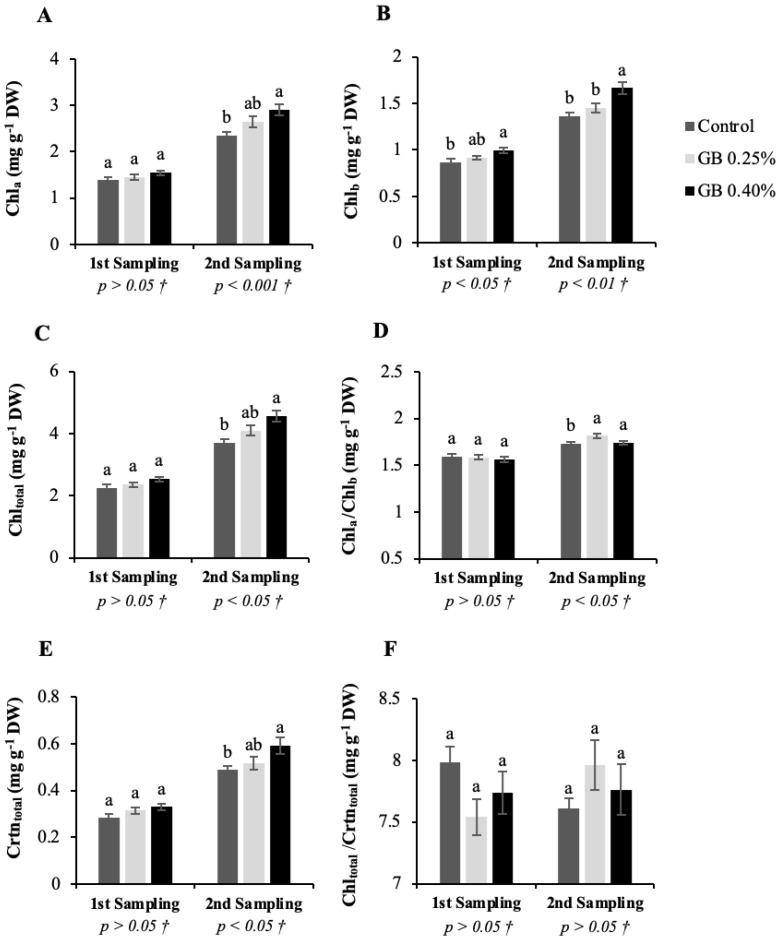
Represents photosynthetic pigments in sweet cherry tree leaves. Chlorophyll a (Chl_a_) (**A**), chlorophyll b (Chl_b_) (**B**), total chlorophyll (Chl_total_) (**C**), Chl_a_/Chl_b_ (**D**), total carotenoids (Crtn_total_) (**E**), Chl_total/_Crtn_total_ (**F**), for the cv. *Lapins* leaves subjected to the different treatments. † Indicates that the *p* values were calculated from the one-way ANOVA analysis after verifying the assumptions of normality and homogeneity of variances. When a significant effect was observed (*p* < 0.05), Tukey’s post-hoc test was applied.

**Figure 5 plants-11-03470-f005:**
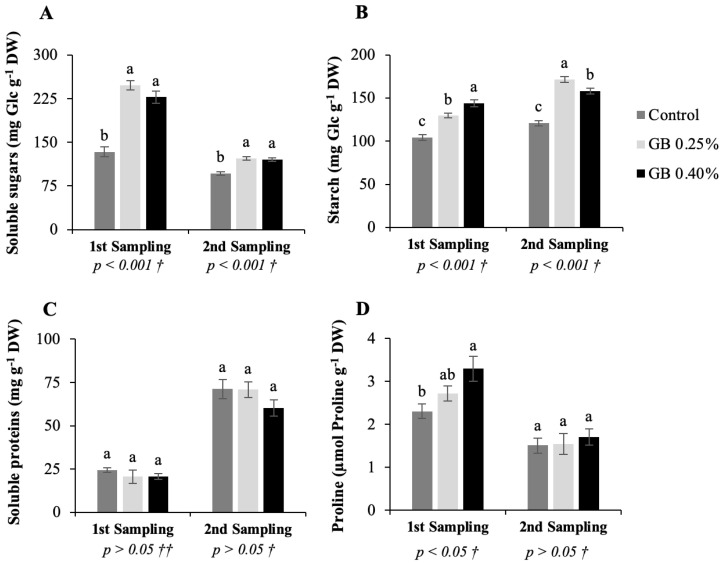
Represents soluble sugars, starch, soluble proteins, and proline concentrations in sweet cherry tree leaves. Soluble sugars (**A**), starch (**B**), soluble proteins (**C**), and proline (**D**) contents, for the cv. *Lapins* leaves subjected to the different treatments. † Indicates that the *p* values were calculated from the one-way ANOVA analysis after verifying the assumptions of normality and homogeneity of variances. When a significant effect was observed (*p* < 0.05), Tukey’s post-hoc test was applied. †† The p values were calculated from the One-way Welch Analysis, since the homogeneity of variances was not verified. When a significant effect was observed (*p* < 0.05), the Dunnett T3’s test was applied.

**Figure 6 plants-11-03470-f006:**
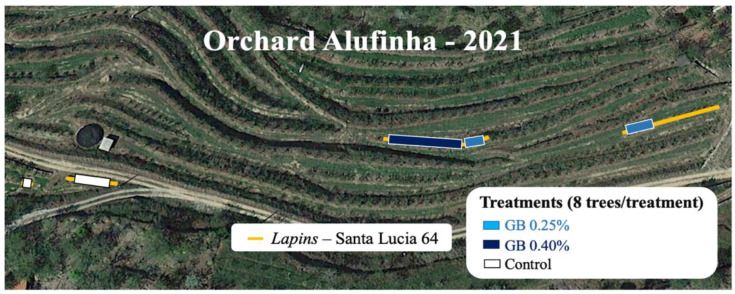
Commercial sweet cherry orchard in Alufinha, Resende.

**Figure 7 plants-11-03470-f007:**
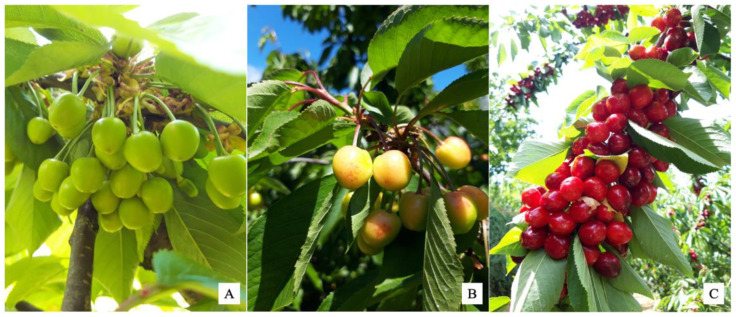
Phenological stages in which treatments were carried out on the sweet cherry trees cv. *Lapins*. 77 BBCH (**A**); 81 BBCH (**B**); and 86 BBCH (**C**).

## Data Availability

Data will be made available on personalized requests due to restrictions from the parent organization.

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
