# Peer review of "Exogenous Application of Glycine Betaine on Sweet Cherry Tree (Prunus avium L.): Effects on Tree Physiology and Leaf Properties"

_plants, 2022, doi:10.3390/plants11243470_

Round 1

Reviewer 1 Report

The manuscript title “Exogenous application of glycine betaine on sweet cherry tree (Prunus avium L.): effects on tree physiology and leaf properties” is an excellent study and have scientific worth. The abstract, introduction and material are written well but results and discussion section have some problems. This MS need major improvement specially the ABTS assay needs to repeat; my detail suggests for authors are as follows.

Reviewer Comments:

1-      In each figure common figure title is missing. Don’t directly explain the sub-figures A, B, and C. Please give one common title than one by one explain: for example:

Figure 7: Represents the photosynthetic pigments in the sweet cherry leaves. Chla (A), Chlb (B), Chltotal (C), Chla/Chlb (D), Crtntotal (E), Chltotal/ 384 Crtntotal (F), for the cv. Lapins leaves subjected to the different treatments.

2-      Figure 3 has three sub-figures, A, B, C, but author forget to cite the figure 3A, figure 3B, and figure 3C. Authors divided each figure into sub-figures; but in text they just cited as figure 4; figure 5…. etc. This is wrong. You have to cite each sub-figure in the text.

3-      Line 323: “The concentration of these compounds increased with leaf age, therefore 2nd sampling analysis revealed higher total phenolic contents, with a range between 53.91 and 72.07 mg GAE g-1 DW (Anwar et al., 2017).” Authors explaining their own research results here, than why they cited “Anwar et al., 2017” at the end of sentence??? Does it make sense; the reader will think its other paper results not from this research…… Authors have to cite their own figure here such as (Figure 6A).

4-      Remove the figure 4 from Manuscript, and move it to supplementary file.

5-      Line 333: the author said “In both samplings, the ABTS method did not reveal significant differences” however, the phenolic contents were increased from 37.23 to 72.07 mg GAE g-1 DW (Figure 6A). The total phenolics contents were almost doubled whereas the ABTS didn’t changed???? According to previously published paper “increase in phenolic contents will increase the ABTS” https://doi.org/10.1186/1756-0500-7-560. Please repeat this antioxidant assay. Also repeat the DPPH assay, because the phenolics were too high in the second sampling whereas the DPPH values have little increment.

6-      As the authors conducted ANOVA, I suggest authors to provide ANOVA table.

7-      Line 339-340. It is antioxidant capacity, not activity.

Author Response

Dear Reviewer,

We sincerely thank you for handling the reviewing of our manuscript. All the comments and suggestions allowed us to improve standard of the manuscript.

In this response, we have taken all comments into consideration and addressed each one individually. As a result, we made some changes in the manuscript, to accommodate the recommendations of the reviewers.

Sincerely,

Marta Serapicos

Response to Reviewer 1 Comments

Point 1: In each figure common figure title is missing. Don’t directly explain the sub-figures A, B, and C. Please give one common title than one by one explain: for example:

Figure 7: Represents the photosynthetic pigments in the sweet cherry leaves. Chla (A), Chlb (B), Chltotal (C), Chla/Chlb (D), Crtntotal (E), Chltotal/ 384 Crtntotal (F), for the cv. Lapins leaves subjected to the different treatments.

Response 1: Thanks for your suggestion. Common titles added to the figures.

Point 2: Figure 3 has three sub-figures, A, B, C, but author forget to cite the figure 3A, figure 3B, and figure 3C. Authors divided each figure into sub-figures; but in text they just cited as figure 4; figure 5…. etc. This is wrong. You have to cite each sub-figure in the text.

Response 2: Thanks for this correction, sub-figures are now cited.

Point 3: Line 323: “The concentration of these compounds increased with leaf age, therefore 2nd sampling analysis revealed higher total phenolic contents, with a range between 53.91 and 72.07 mg GAE g-1 DW (Anwar et al., 2017).” Authors explaining their own research results here, than why they cited “Anwar et al., 2017” at the end of sentence??? Does it make sense; the reader will think its other paper results not from this research…… Authors have to cite their own figure here such as (Figure 6A).

Response 3: Thanks for your correction. Citation “Anwar et al., 2017” was moved to the correct place.

Point 4: Remove the figure 4 from Manuscript and move it to supplementary file.

Response 4: Thanks for your suggestion. Figure 4 was moved and renamed (Figure S1. Mean air temperature (°C) from January to June of 2021.). Should we add a section to our paper called “Supplementary Materials” or send an attachment as a supplementary file?

Point 5: Line 333: the author said “In both samplings, the ABTS method did not reveal significant differences” however, the phenolic contents were increased from 37.23 to 72.07 mg GAE g-1 DW (Figure 6A). The total phenolics contents were almost doubled whereas the ABTS didn’t changed???? According to previously published paper “increase in phenolic contents will increase the ABTS” https://doi.org/10.1186/1756-0500-7-560. Please repeat this antioxidant assay. Also repeat the DPPH assay, because the phenolics were too high in the second sampling whereas the DPPH values have little increment.

Response 5: Many thanks for your attention and comment. All antioxidant activity assays were at least performed two times and the results were always similar. At this point the only thing we could do is remove these results from the article.

Point 6: As the authors conducted ANOVA, I suggest authors to provide ANOVA table.

Response 6: Thanks for your suggestion, we completely agree with you, however it’s not possible to provide ANOVA.

Point 7: Line 339-340. It is antioxidant capacity, not activity.

Response 7: Thanks, correction performed.

Reviewer 2 Report

The paper is well written, although I have added many grammatical corrections for there authors to incorporate as they see fit. There were two issues that the authors have to correct.

1) the experimental design is pseudo replicated. Unfortunately the subsamples (n=8) within a treatment are NOT true replicates because they are clustered together in geographic blocks. Therefore, it is not possible to determine if the experimental effects observed are due to treatment or due to the geographic location of the trees in the orchard. I do not think that the study should be rejected because the authors still supply useful information, but they do HAVE to state that their experiment was pseudo replicated and thus conclusions have to be considered with caution. If the authors do not provide this explanation in the manuscript then the paper should NOT be published. Technically ANOVA or any other statistical analysis is not suitable for pseudo replicated designs, just graphs and tables without statistical inference is fine. However, in this case I feel that it does no harm to include the results of the ANOVA if the authors explain in the statistics section of their methods that they performed ANOVA anyhow, despite the pseudoreplication.

2. In Figure 3A and Figure 7B have mistakes in the assignment of the letters denoting differences between treatment means... please see comments in manuscript for details.  

Author Response

Dear Reviewer,

We sincerely thank you for handling the reviewing of our manuscript. All the comments and suggestions allowed us to improve standard of the manuscript.

In this response, we have taken all comments into consideration and addressed each one individually. As a result, we made some changes in the manuscript, to accommodate the recommendations of the reviewers.

Sincerely,

Marta Serapicos

Response to Reviewer 2 Comments

Point 1: the experimental design is pseudo replicated. Unfortunately the subsamples (n=8) within a treatment are NOT true replicates because they are clustered together in geographic blocks. Therefore, it is not possible to determine if the experimental effects observed are due to treatment or due to the geographic location of the trees in the orchard. I do not think that the study should be rejected because the authors still supply useful information, but they do HAVE to state that their experiment was pseudo replicated and thus conclusions have to be considered with caution. If the authors do not provide this explanation in the manuscript then the paper should NOT be published. Technically ANOVA or any other statistical analysis is not suitable for pseudo replicated designs, just graphs and tables without statistical inference is fine. However, in this case I feel that it does no harm to include the results of the ANOVA if the authors explain in the statistics section of their methods that they performed ANOVA anyhow, despite the pseudoreplication.

Response 1: We appreciate the reviewer's comment and agree with it. However, given that the commercial orchard was very homogeneous in terms of soil type, slope and selected trees, each tree was assumed to be a repetition, making it impossible to carry out the treatments (foliar application of the compounds) in any other way. The phrase 'Since the commercial orchard was very homogeneous in terms of soil type, slope and selected trees, each tree was assumed to be a repetition,' was introduced into the manuscript.

Point 2: In Figure 3A and Figure 7B have mistakes in the assignment of the letters denoting differences between treatment means... please see comments in manuscript for details. 

Response 2: Thanks for your attention and correction, correction performed.

Reviewer 3 Report

- This manuscript correspond for scope of journal.

- Content of article apropriate to the title of article 

- Key words are appropriate. 

- Methods of investigation are adequate. 

- Results are clearly presented and discussion. 

- Table, figure and picture are clear.

- Conclusion are based on obtained results, but last sentence need improved and need write conclusion sentence without next part "Although this study did not address abiotic stress" 

REMRKS: 

In line 332 the sentence is not cler!? - likely it is continuance of sentence which begin in line 324 ("Regarding .... phenolic, there ...."). My observation is that this confusion was caused by the insertion of a paragraph from line 329 to 331. Please correct that. 

Author Response

Dear Reviewer,

We sincerely thank you for handling the reviewing of our manuscript. All the comments and suggestions allowed us to improve standard of the manuscript.

In this response, we have taken all comments into consideration and addressed each one individually. As a result, we made some changes in the manuscript, to accommodate the recommendations of the reviewers.

Sincerely,

Marta Serapicos

Response to Reviewer 3 Comments

Point 1:

- This manuscript correspond for scope of journal.

- Content of article apropriate to the title of article 

- Key words are appropriate. 

- Methods of investigation are adequate. 

- Results are clearly presented and discussion. 

- Table, figure and picture are clear.

- Conclusion are based on obtained results, but last sentence need improved and need write conclusion sentence without next part "Although this study did not address abiotic stress" 

REMRKS: 

In line 332 the sentence is not cler!? - likely it is continuance of sentence which begin in line 324 ("Regarding .... phenolic, there ...."). My observation is that this confusion was caused by the insertion of a paragraph from line 329 to 331. Please correct that. 

Response 1: Conclusion rearranged. 332 paragraph rearranged.

Round 2

Reviewer 1 Report

Comments for authors:

The authors revised the MS, however, ot need minor improvements regarding reference citations.

Line 36, 43, 49, and so on..... the references citation parenthesis should placed at appropiate place according to the jounal guidlines.